# Confocal Families of Hyperbolic Conics via Quadratic Differentials

**Joel Langer** †  and **David Singer** *,†

Department of Mathematics, Applied Math and Statistics, Case Western Reserve University, Cleveland, OH 44106-7058, USA; joel.langer@case.edu
* Correspondence: david.singer@case.edu
† These authors contributed equally to this work.

**Abstract:** We apply the theory of quadratic differentials, to present a classification of orthogonal pairs of foliations of the hyperbolic plane by hyperbolic conics. Light rays are represented by trajectories of meromorphic differentials, and mirrors are represented by trajectories of the quadratic differential that represents the geometric mean of two such differentials. Using the notion of a hyperbolic conic as a mirror, we classify the types of orthogonal pairs of foliations of the hyperbolic plane by confocal conics. Up to diffeomorphism, there are nine types: three of these types admit one parameter up to isometry; the remaining six types are unique up to isometry. The families include all possible hyperbolic conics.

**Keywords:** quadratic differential; hyperbolic conic; Poincaré disc; confocal families

**MSC:** 51M10; 30F30





## 1. Introduction

In this article we apply the theory of quadratic differentials, to present a classification of orthogonal pairs of foliations of the hyperbolic plane by hyperbolic conics. This complements the classifications in the Euclidean and spherical cases that appear in [1]. In that paper, which was a useful reference for the current discussion, we viewed conics as mirrors reflecting a field of geodesics, representing a bundle of light rays, to another field of geodesics. The light rays were trajectories of meromorphic differentials; the mirrors were trajectories of the quadratic differential that represents the geometric mean of the two families.

Here, we demonstrate that the curves we describe as mirrors are the hyperbolic conics described by various authors, e.g., [2–6], using projective geometry (we neglect curves with an empty locus in the hyperbolic plane). In fact, every such hyperbolic conic gives rise to an orthogonal pair of confocal families of conics, given by the horizontal and vertical trajectories of a quadratic differential.

As we noted in [1], the hyperbolic case is considerably more complicated than the Euclidean and spherical cases, because of the wider variety of geodesic fields. In addition to families of rays emanating from a point, and families of "parallel" rays converging to an ideal point, there are also "ultraparallel" (some authors use the term "hyperparallel" for lines that diverge in both directions) families that in some models of hyperbolic geometry may be thought of as converging to an "ultra-ideal" point. This leads to a more elaborate classification of conics: in addition to the obvious notions of ellipse, parabola, and hyperbola, there are curves such as semihyperbolas. As we will see, such curves may be viewed as reflecting light from a point source (a focus) to a hyperparallel field of geodesics: namely, the orthogonals to a fixed hyperbolic line.

We may visualize the semihyperbolic mirror in the Poincaré disc model of hyperbolic space or in the Beltrami–Klein model of the disc (Figure 1). Each model has advantages and disadvantages. In the Poincaré model, the light rays are circle arcs meeting the circle

orthogonally, or straight lines from the origin: it is clear that the angle of incidence equals the angle of reflection. The reflected rays are orthogonal to a hyperbolic line, shown as the dotted curve, and are seen to diverge from each other. In the Klein model, the light rays are all straight lines. The rays converge to a point beyond the ideal boundary (Figure 2, right). In this model, we can view the semihyperbola as having two foci. Viewing the Klein disc as part of the Euclidean plane, the curve is seen to be a Euclidean ellipse, although one of its foci is actually outside the ellipse. It is not obvious that the picture is correct, as the Klein model is not conformal.

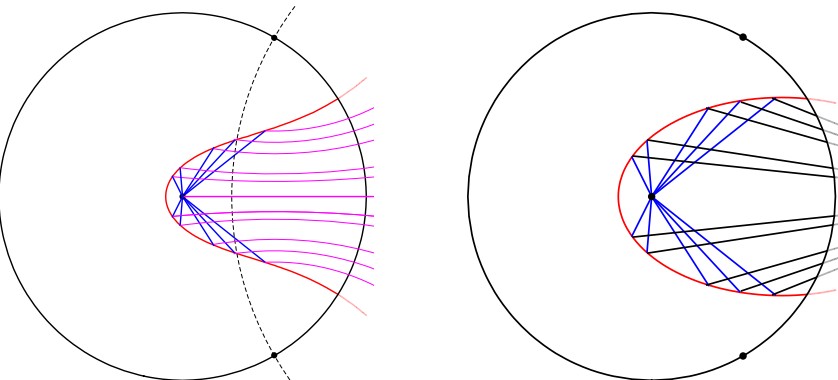

**Figure 1.** Semihyperbolic mirror in Poincaré disc (**left**); in Klein disc (**right**). The pink arcs in the figure on the left represent light rays, which are reflected off the (orange) mirror to the blue light rays that converge at the focus. In the Klein model, these rays are represented by black straight lines.

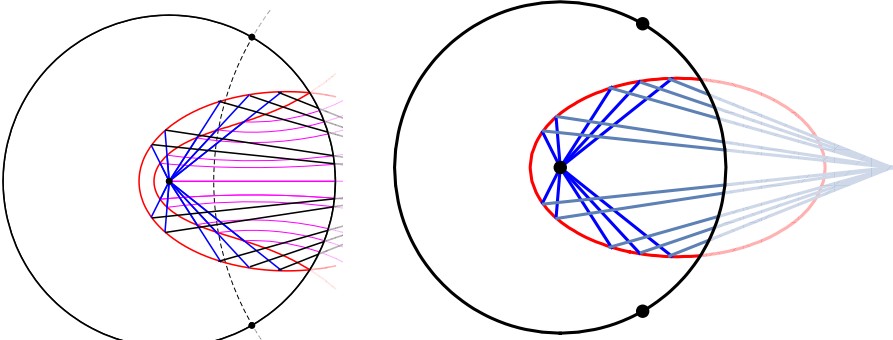

**Figure 2.** **Left**: The two models superimposed. **Right**: Locating the ultra-ideal point in the Klein model. Red = mirror; pink and gray = incident rays, blue = reflected rays.

Perhaps the best way to see that the curves in the Klein model are correct is to compare the two pictures via the Klein-to-Poincaré map $KP(z) = \frac{z}{1+\sqrt{1-z\bar{z}}}, |z| \leq 1$: this is the unique isometry from the disc with the Klein metric to the disc with the Poincaré metric that keeps the ideal boundary pointwise fixed [7]; it takes the straight line segment between points on the boundary to the geodesic between the same two points in the Poincaré disc. As we have chosen the focus inside the disc to be at the origin, the rays from the focus coincide (but not pointwise). The reflected rays meet the ideal boundary at the same point in the two models (Figure 2, left).

There is no relation between points outside the disc in the two models. The Klein disc can be thought of as a subset of the projective plane $\mathbb{R}P^2$. This idea apparently dates back to a paper by Study [8]. Some authors refer to this as the "extended hyperbolic plane", e.g., [4,9]. The exterior of the disc is the "de Sitter Plane". The Poincaré disc, on the other hand, is a subset of the extended complex plane $\mathbb{C}P^1$. In this model, points in the exterior of the unit disc may be thought of as being identified with points in the interior via reflection across the unit circle.

To study confocal families of conics, we use the theory of quadratic differentials. We will see that a family of confocal hyperbolic conics can be represented in the Poincaré disc model as the horizontal or vertical trajectories of a non-vanishing meromorphic quadratic differential: this is consistent with viewing hyperbolic conics as parts of curves in $\mathbb{CP}^1$. They may instead be seen as certain quartic or lower degree curves in $\mathbb{CP}^2$, though we will not make use of that fact here.

In Section 2, we discuss the general notion of confocal families of conics, and of the bundles of light rays reflected by a confocal family. Section 3 reviews the fields of geodesics and the corresponding differentials used in constructing the families. Then, in Section 4, we enumerate the quadratic differentials determined by the geodesic fields and nine types of confocal families of conics. Appendix A gives a dictionary of the types of conics as they are defined by various authors, and where to find them among the confocal families. Appendix B has some comments on how the curve families are computed. For background material on quadratic differentials, refer to [10] or [1].

## 2. Confocal Conics in the Hyperbolic Plane

In order to discuss confocal conics, it will be necessary to determine what we mean by the term "confocal". In the case of the ellipse, the concept dates back at least to the work of Menaechmus (c. 350 BCE), appearing in the book *Conics* by Apollonius of Perga. While he does not name the points, Apollonius showed that the foci of an ellipse satisfy two important properties: the string property and the reflection property. It was Kepler who introduced the term foci, noting that the orbits of the planets are ellipses, with the sun located at one of the foci.

The two properties of ellipses continue to hold in the sphere and the hyperbolic plane: that is, the sum of the distances of a point on the ellipse to the foci is constant, and the lines joining the point to the foci make equal angles with the tangent line. It follows that light rays emanating from one focus, and reflected off the ellipse, travel to the other focus. It is thus clear what we mean by confocal ellipses.

For parabolas, the notion of confocal is already more subtle in the Euclidean case. As a parabola only has one focus in the plane, a rotation around the focus carries a parabola to another parabola with the same focus. Thus, to define confocal parabolas, we must take into account the direction determined by parallel rays that are reflected by the parabola into its focus. A natural way to understand this is to think of the parabola as having a second focus at infinity. If we view the plane as being a subset of $\mathbb{RP}^2$, then an ordinary point and an ideal point determine two confocal families of parabolas, one orthogonal to the other. This idea needs to be taken a step further when we consider the hyperbolic case, where a focus may be located at infinity or even beyond. In the Klein model, the foci of a conic may be points in the hyperbolic plane, ideal points, or de Sitter points ([4], p. 30).

**Remark 1.** *In [3], Henle discusses two conditions that are equivalent definitions of Euclidean parabolas. He shows that the curves in $\mathbb{H}$ that are equidistant from a point (focus) and a line (directrix) are not the same as the curves that reflect rays from a focus to an ideal point. His computations allow one to verify that the curves satisfying the focus–directrix condition are the semihyperbolas, while the curves satisfying the reflection property are the elliptic parabolas, which may be thought of as the limiting case of semihyperbolas, as the directrix becomes infinitely distant (Figure 1).*

The notion that a focus may be located at an ultra-ideal point may be justified by considering a variable mirror with one fixed focus at the origin in the disc, while the other focus is allowed to move across the ideal boundary, as seen in Figures 3 and 4. When the movable focus reaches the ideal boundary, the reflected light rays become parallel, in the hyperbolic sense, i.e., they meet at the ideal point. If we "keep going" beyond the ideal boundary, the reflected rays now disperse. Note that the mirror has only one branch in the hyperbolic plane; hence, the name "semihyperbola". The same picture in the Klein

model would show the rays meeting at the ultra-ideal focus, as we saw in Figure 2. In the extended hyperbolic plane, this process is continuous.

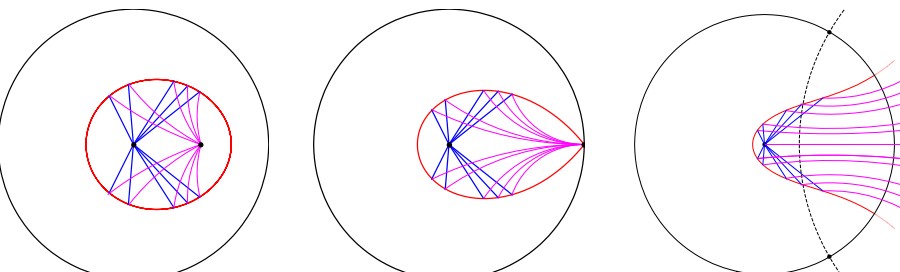

**Figure 3.** One focus at the origin, the other moving right, ellipses to parabolas to semihyperbolas, in the Poincaré disc model. Red = mirror; pink = incident rays, blue = reflected rays.

We regard a conic in the hyperbolic plane $\mathbb{H}$ (normalized to have constant Gaussian curvature $-1$) as a mirror, which reflects a light ray according to the principle that angle of incidence equals angle of reflection. The mirror has a coherence property: there is a particular bundle of light rays $I$ that is reflected into another particular bundle $J$. These bundles of rays are oriented geodesic fields whose orthogonal trajectories have constant geodesic curvature. In the Euclidean plane, this condition is satisfied by the rays emanating from a point, and by the parallel lines orthogonal to a fixed line.

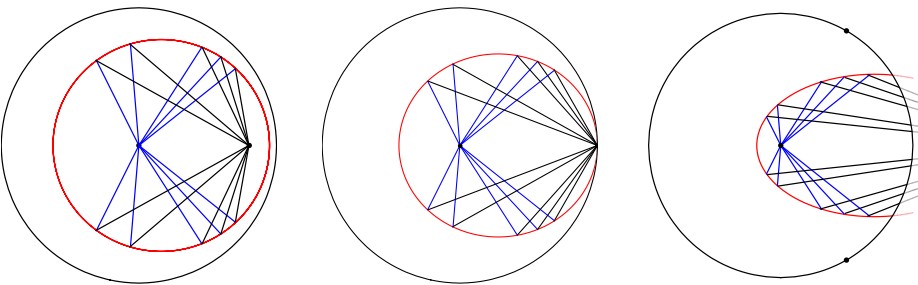

**Figure 4.** One focus at the origin, the other moving right, in the Klein model. Red = mirror; black = incident rays, blue = reflected rays.

In the hyperbolic plane, there are three types of such fields. An elliptic geodesic field consists of geodesic rays emanating from (or converging to) a point in $\mathbb{H}$; its orthogonal trajectories are circles (curves of constant curvature $> 1$) (Figure 5). A hyperbolic geodesic field consists of rays orthogonal to a fixed geodesic (Figure 6, right): such a field's other orthogonal trajectories have constant curvature $0 < k < 1$. The transition between these two types consists of rays asymptotic to an ideal point. For such a parabolic geodesic field, the orthogonal trajectories are horocycles, which have curvature $k = 1$ (Figure 6, left).

We refer to the above special geodesic fields collectively as elliptic–parabolic–hyperbolic fields, or EPH fields. For any EPH field, if we take the unit vector field $T$ tangent to the family of rays, the flow determined by integrating $T$ preserves the orthogonal trajectories, which we may think of as wave fronts. This fact is actually a special case of Gauss' lemma in Riemannian geometry, and it characterizes geodesic fields.

**Definition 1.** *A confocal family of conics is a family of curves that reflects some fixed EPH field $I$ to another fixed EPH field $J$.*

This notion of confocal family has the advantage that it is not model-dependent. In the Klein model, conics are recognizable as being Euclidean conics, a fact that is often exploited in discussions of hyperbolic geometry. One can "see" the foci as points in the disc, on its boundary, or outside the disc; however, as this model is not conformal, it is difficult to determine the

location of foci for a given conic, or to recognize when conics are confocal. We will be relying mainly on the Poincaré disc model, which is conformal, and allows us to use complex analysis: in this model, however, the conics are quartic (or lower degree) curves in $\mathbb{C} \simeq \mathbb{R}^2$, and the ultra-ideal points are not points in the model; instead, the geodesic fields are represented by pencils of circles orthogonal to the unit circle. This representation of geodesic fields, expressed in terms of differentials, forms the basis for the procedure followed in the next section.

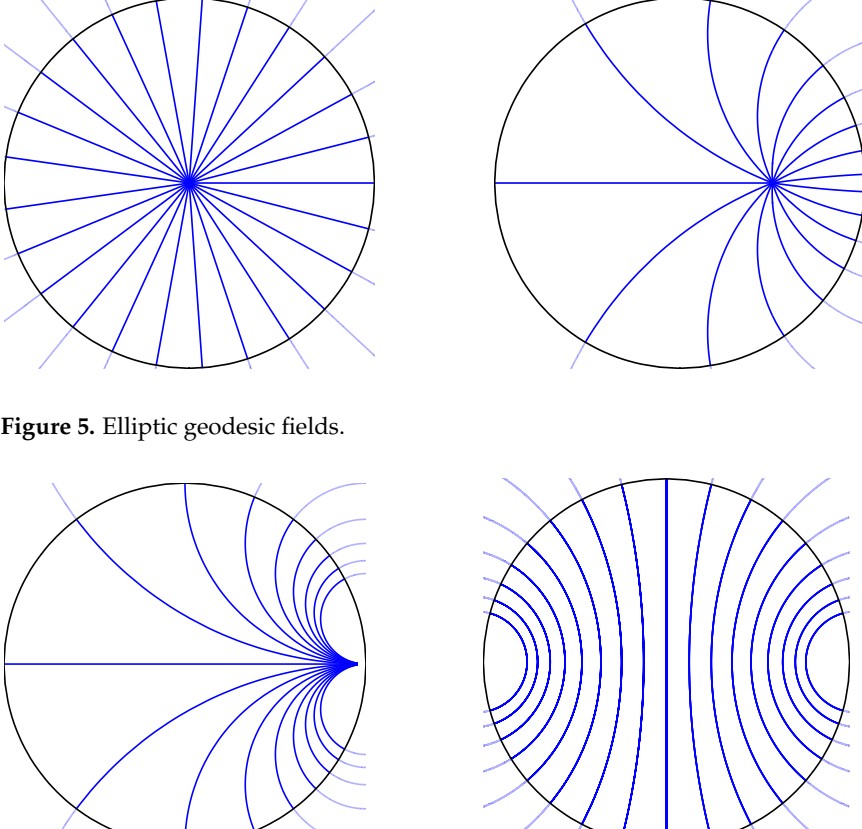

**Figure 5.** Elliptic geodesic fields.

**Figure 6.** Parabolic geodesic fields (**left**) and hyperbolic geodesic fields (**right**).

## 3. Curve Families in the Disc and Meromorphic Differentials

From here on, we use the Poincaré disc model $\mathbb{D} \simeq \mathbb{H}$ of the hyperbolic plane. As $\mathbb{D}$ is the open unit disc in the extended complex plane $\hat{\mathbb{C}} \simeq \mathbb{CP}^1$, we may represent curve families in $\mathbb{D}$ via trajectories of meromorphic differentials on $\hat{\mathbb{C}}$. An EPH field is represented by the (oriented) trajectories of a differential $f(z)dz$. Given two such EPH fields, the corresponding confocal conics are represented via the product quadratic differential $Q = f_1(z)f_2(z)dz^2$. The (unoriented) horizontal and vertical trajectories of $Q$ form a pair of orthogonal curve families bisecting the angle between the pair of geodesic fields: thus, such trajectories act as mirrors reflecting one EPH field to the other.

**Remark 2.** *The resulting curve families can be transferred to the Klein model, using the Klein-to-Poincaré and Poincaré-to-Klein maps [7]. The quartic curves in the disc $\mathbb{D} \subset \mathbb{C}$ are transferred to quadratic curves in the Klein disc model, allowing us to see what confocal conics look like in the latter model.*

In this section, we describe the differentials $f(z)dz$ representing the three types of EPH fields. Then, the classification of confocal families of hyperbolic conics, in the next section, amounts to combining pairs of such differentials in all possible ways, up to hyperbolic isometries (i.e., conformal automorphisms of $\mathbb{D}$).

Although points in the exterior of $\mathbb{D} \subset \hat{\mathbb{C}}$ are generally ignored in the Poincaré disc model, it is useful to consider the circle inversion $z \mapsto \mathcal{I}(z) := 1/\bar{z}$, which interchanges $\mathbb{D}$ with its exterior. The geodesics in $\mathbb{D}$ are represented by circle arcs meeting the ideal boundary $S^1 = \partial\mathbb{D}$ orthogonally, and one may view inversion as *geodesic-reversing*. Accordingly, the differential $\omega = f(z)dz$ corresponding to an EPH field transforms under inversion to its opposite $\omega \mapsto -\omega$. (Recall that trajectories $\gamma(t)$ of $\omega$ are defined by the positivity condition $f(\gamma(t))\gamma'(t) > 0$, so reversing the sign of $\omega$ reverses orientation of trajectories.)

In particular, the singular set of such a differential $\omega$ must be preserved by $\mathcal{I}$: that is, a singularity either belongs to an inversion-symmetric pair $\{a, 1/\bar{a}\}$, $a \in \mathbb{D}$, or is a complex unit $\sigma \in S^1$. Furthermore, from the description of the three types of EPH fields, it is now evident that $\omega$ has, at most, two singularities in $\hat{\mathbb{C}}$. As the zeros and poles of a meromorphic differential on $\hat{\mathbb{C}}$ satisfy $Z - P = 2g - 2 = -2$, it follows that $Z = 0$ and $P = 2$, i.e., $\omega$ is non-vanishing, and has either two simple poles or one double pole. In fact, $\omega$ has either a pair of simple poles $a, 1/\bar{a}$, for some $a \in \mathbb{D}$, a double pole $\sigma = e^{i\alpha} \in S^1$, or a pair of simple poles $\sigma_j = e^{i\alpha_j} \in S^1$. Thus, $\omega$ has one of the three forms:

(1)　　$\omega = \dfrac{\lambda dz}{(z-a)(z-1/\bar{a})}$, $a \in \mathbb{D}$;

(2)　　$\omega = \dfrac{\lambda dz}{(z-\sigma)^2}$, $\sigma = e^{i\alpha} \in S^1$; or

(3)　　$\omega = \dfrac{\lambda dz}{(z-\sigma_1)(z-\sigma_2)}$, $\sigma_j = e^{i\alpha_j} \in S^1$.

Here, it remains to determine values of the *multipliers* $\lambda = re^{i\theta} \in \mathbb{C} \setminus \{0\}$, such that $\omega \mapsto -\omega$ under inversion.

For this purpose, it is useful to first consider the reflection of rational functions and differentials in the real axis. For a function, this is given by $\bar{f}(z) := \overline{f(\bar{z})}$ (complex conjugation of coefficients). In particular, for $f$ to be reflection anti-symmetric, $\bar{f} = -f$, it must have imaginary coefficients. The reflection of a differential $\omega = f(z)dz$ in the real axis is given by $\bar{\omega} = R_{\mathbb{R}}\omega := \bar{f}(z)dz$. In the anti-symmetric case, $\bar{\omega} = -\omega$, $f$ must again have imaginary coefficients.

The corresponding rule for reflecting differentials $\omega = f(z)dz$ in the unit circle can now be described via pull-back, by $S(z) = 1/z$: namely, $R_{S^1}\omega := S^*\bar{\omega} = -\dfrac{\bar{f}(1/z)dz}{z^2}$. For differentials of the form $\omega = \dfrac{\lambda dz}{g(z)}$, as above, the rule for reflection anti-symmetry $-\omega = R_{S^1}\omega$ becomes $\dfrac{\lambda}{g(z)} = \dfrac{\bar{\lambda}}{z^2\bar{g}(1/z)}$, i.e., $e^{2i\theta} = \lambda/\bar{\lambda} = \dfrac{g(z)}{z^2\bar{g}(1/z)}$. Thus, $g$ determines $e^{i\theta}$, up to sign, via the formula $e^{2i\theta} = \dfrac{g(z)}{z^2\bar{g}(1/z)}$. For instance, in case (1), $e^{2i\theta} = \dfrac{(z-a)(z-1/\bar{a})}{z^2(1/z-\bar{a})(1/z-1/a)} = a/\bar{a}$; therefore, up to multiplication by a positive number, $\omega = \dfrac{adz}{(z-a)(z-1/\bar{a})}$. In case (2), $e^{2i\theta} = \dfrac{(z-\alpha)^2}{z^2(1/z-\bar{\alpha})^2} = \alpha/\bar{\alpha}$, so we can take $\omega = \dfrac{\alpha dz}{(z-\alpha)^2}$.

Case (3) is slightly more complicated, in general; however, for our purposes, it will suffice to symmetrize $\omega$, so that $\sigma_1$ and $\sigma_2$ are complex conjugates: $e^{2i\theta} = \dfrac{(z-\sigma)(z-\bar{\sigma})}{z^2(1/z-\bar{\sigma})(1/z-\sigma)} = \dfrac{(z-\sigma)(z-\bar{\sigma})}{(1-\bar{\sigma}z)(1-\sigma z)} = 1$. Thus, we can simply take $\omega = \dfrac{\pm dz}{(z-\sigma)(z-\bar{\sigma})}$.

Of course, any given differential in case (3) may be put into this symmetrized form via rotation; however, as we will be considering pairs of differentials, more general hyperbolic isometries (disc automorphisms) are actually required, to show that such symmetrized differentials suffice. Similarly, for the first two types of differentials, it turns out that we may always take $\lambda = \pm 1$: in case (1), we may always assume $-1 < a < 1$, and in case (2), we may use $\sigma = \pm 1$.

It is not necessary to consider all possible ways to multiply two differentials $\omega_1, \omega_2$ of the above types, because we can characterize the resulting quadratic differentials $Q = q(z)dz^2 = \omega_1\omega_2$ more directly. First, we note that the equations $R_{S^1}\omega_j = -\omega_j$ im-

ply that $Q$ itself has reflection symmetry $R_{S^1}Q = Q$. $(R_{S^1}Q = S^*\bar{Q} = (R_{S^1}\omega_1)(R_{S^1}\omega_2))$. Counting multiplicities, $Q$ has exactly four poles, and these are either paired by reflection in $S^1$, or lie on $S^1$. Furthermore, by hyperbolic isometries, we can arrange for all poles to be real or belong to complex conjugate pairs. Figure 7 shows the resulting possible pole patterns. The nine types I–IX of quadratic differentials (pole patterns) are discussed further in Section 4.

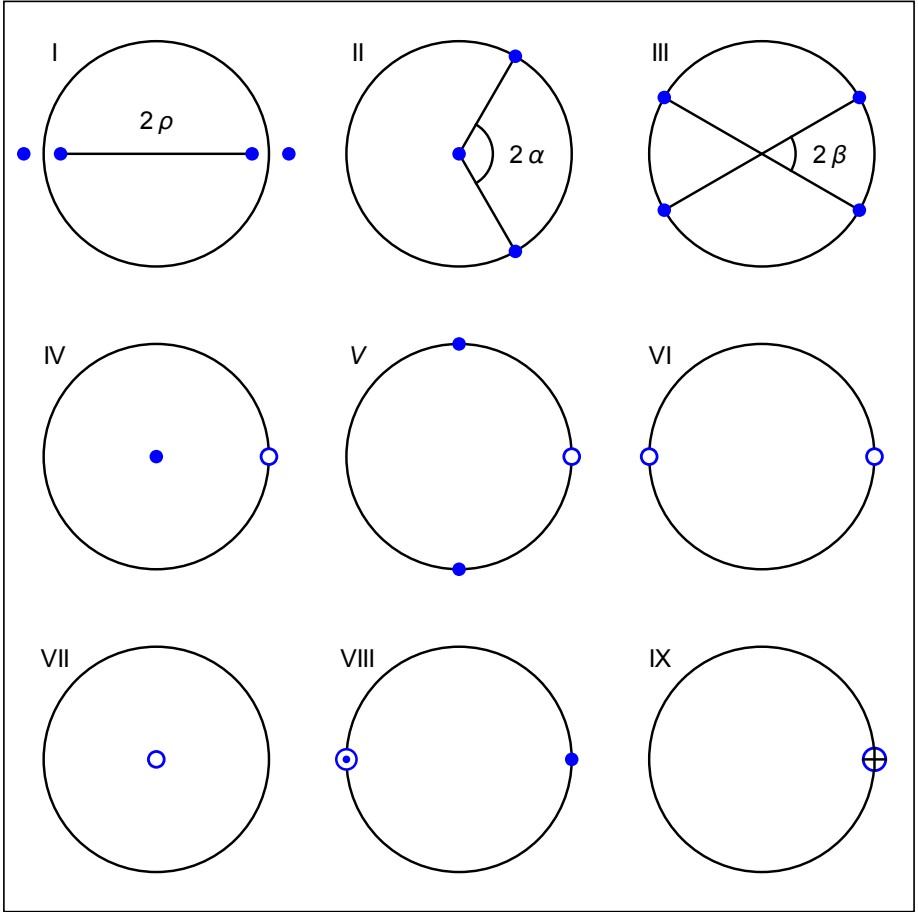

**Figure 7.** Pole patterns for the quadratic differentials. Pole key: $\bullet$ = simple; $\circ$ = double; $\odot$ = triple; $\oplus$ = quadruple. Poles at $z = 0$ are paired with identical poles at $z = \infty$—out of view. Hyperbolic–geometric parameters for types I, II, and III: pole separation $0 < 2\rho < \infty$; angle between geodesic rays $0 < 2\alpha < \pi$; angle between geodesics $0 < 2\beta \leq \pi/2$.

For example, Type I arises when there are four poles of the form $a, 1/\bar{a}, b, 1/\bar{b}, |a| < 1$, $|b| < 1$: in this case, the midpoint of the geodesic arc $\overline{ab} \subset \mathbb{D}$ of length $2\rho$ can be moved to the origin by isometry, and then rotation places its endpoints on the real axis, with equal distance $\rho$ from the origin, as in Figure 7.

For Type II, there are poles $a, 1/\bar{a}, \sigma_1 \neq \sigma_2, |a| < 1, |\sigma_j| = 1$; then, $a$ may be moved to the origin, and rotation gives $\sigma_2 = \bar{\sigma}_1$. Within this type, equivalence is defined by the angle $0 < 2\alpha < \pi$ between geodesic rays from $a$ to $\sigma_1$ and $\sigma_2$.

For Type III, there are four (consecutive) poles $\sigma_j \in S^1$. These can be put in a long rectangular position: geodesics $\overline{\sigma_1\sigma_3}$ and $\overline{\sigma_2\sigma_4}$ may be moved, so as to intersect at $z = 0$; the resulting rectangular pole-set may then be made "horizontal" by rotation (and the angle between the diagonal geodesics then lies in the range $0 < 2\beta \leq \frac{\pi}{2}$). Note that there are three ways to represent a "rectangular" type such as $Q = \omega_1\omega_2$, one of which corresponds to differentials satisfying $\bar{\omega}_j = \omega_j$.

Finally, the "multiplier" in $Q = \frac{\Lambda dz^2}{G(z)} = \frac{\lambda_1\lambda_2 dz^2}{g_1(z)g_2(z)}$ is again determined, up to real scaling, by the poles. Thus, for the normal forms shown in Figure 7, $Q$ is always of the form

$Q = \frac{\pm dz^2}{G(z)}$, where $G(z)$ is a real, monic polynomial of degree—at most, *four*. A change of sign in $Q$ simply interchanges horizontal and vertical trajectories.

## 4. Classification of Confocal Families

For each of the types listed below, we specify the form of the quadratic differential whose horizontal and vertical trajectories give confocal families of conics. We specify the location of the poles. In the first three cases, there is a one-parameter family of isometric classes; in the remaining cases, all the families are isometric. The curves in each family act as mirrors; we describe the geodesic fields that are reflected in these mirrors. Finally, we illustrate the families graphically, in both the Poincaré disc and the Klein disc. The curves in the Klein model are recognizable as parts of Euclidean conics; the corresponding curves in the Poincaré disc are quartics that meet the circle in the same points as the Klein curves. To get from one model to the other, one can use the Poincaré-to-Klein map and its inverse.

(I)  Two poles in the disc, and their reflections in the circle (Figure 8).
  The quadratic differential is equivalent to

$$\frac{dz^2}{(z^2 - a^2)(z^2 - 1/a^2)},$$

with $0 < a < 1$. The trajectories are hyperbolic ellipses and hyperbolas, with foci in the Poincarè disc at $z = a$ and $z = -a$. The rays from one focus are reflected to the rays to the other focus (ellipses) or from the other focus (hyperbolas).

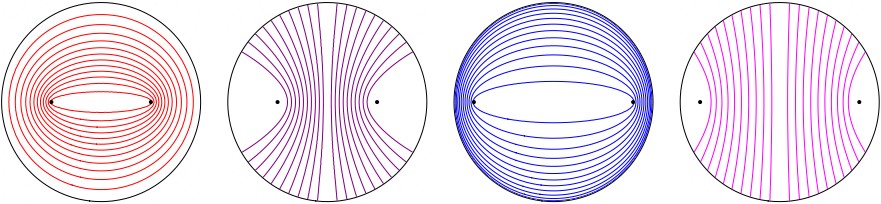

**Figure 8.** I. Ellipse and hyperbola. The left two parts show the horizontal (red) and vertical (magenta) trajectories in the Poincaré disc model; the right two parts show the horizontal (blue) and vertical (pink) trajectories in the Klein model.

(II)  One pole in the disc and its reflection, two poles on the circle (Figure 9).
  The quadratic differential is equivalent to

$$\frac{dz^2}{z(z^2 - 2\cos(\alpha)z + 1)},$$

with $0 < \alpha < \pi/2$. The trajectories are semihyperbolas, with one focus (at the origin). Rays from the focus are reflected to rays corresponding to lines that cross the line joining the ideal points orthogonally.

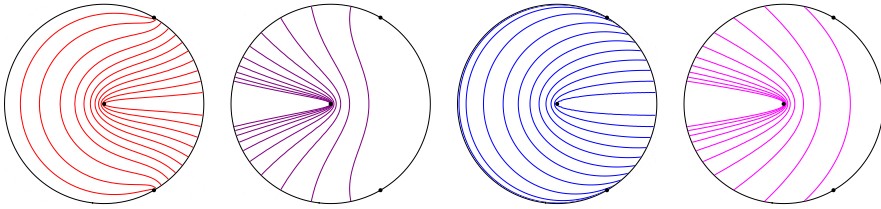

**Figure 9.** II. Semihyperbola. The left two parts show the horizontal (red) and vertical (magenta) trajectories in the Poincaré disc model; the right two parts show the horizontal (blue) and vertical (pink) trajectories in the Klein model.

(III)  Four poles at points on the circle (Figure 10).
The quadratic differential is equivalent to

$$\frac{dz^2}{z^4 - 2\cos(2\beta)z^2 + 1}$$

for $0 < \beta \le \frac{\pi}{4}$. The poles are placed at $\pm e^{i\beta}$ and $\pm e^{-i\beta}$. The trajectories are concave hyperbolas. Rays orthogonal to one line are reflected to rays orthogonal to another line.

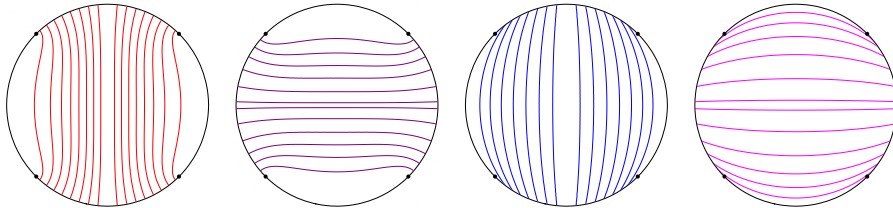

**Figure 10.** III. Concave hyperbolas.The left two parts show the horizontal (red) and vertical (magenta) trajectories in the Poincaré disc model; the right two parts show the horizontal (blue) and vertical (pink) trajectories in the Klein model.

(IV)  A pole in the disc, and its reflection, and a double pole on the circle (Figure 11).
The quadratic differential is equivalent to

$$\frac{dz^2}{z(z-1)^2}.$$

We may place the poles at $z = 0$ and at $z = 1$: this is a limiting case of an ellipse or hyperbola, with one focus at an ideal point. The trajectories are elliptic parabolas, which reflect rays from the finite focus to rays approaching the ideal point, and convex hyperbolic parabolas, which reflect rays from the finite focus to rays emanating from the ideal point.

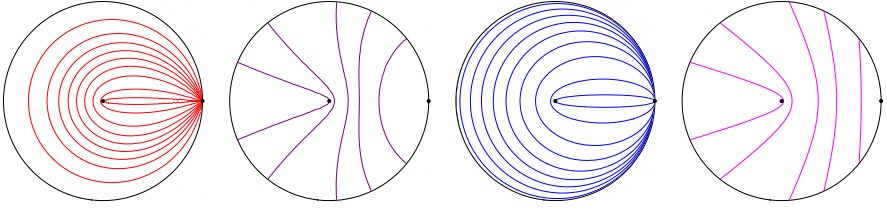

**Figure 11.** IV. Elliptic and convex hyperbolic parabola. The left two parts show the horizontal (red) and vertical (magenta) trajectories in the Poincaré disc model; the right two parts show the horizontal (blue) and vertical (pink) trajectories in the Klein model.

(V)  One double pole and two single poles on the circle (Figure 12).
The quadratic differential is equivalent to

$$\frac{dz^2}{(z-1)^2(z^2+1)}.$$

Rays orthogonal to the line joining the ideal points are reflected to rays to the double ideal point. The mirrors are wide concave hyperbolic parabolas and long concave hyperbolic parabolas.

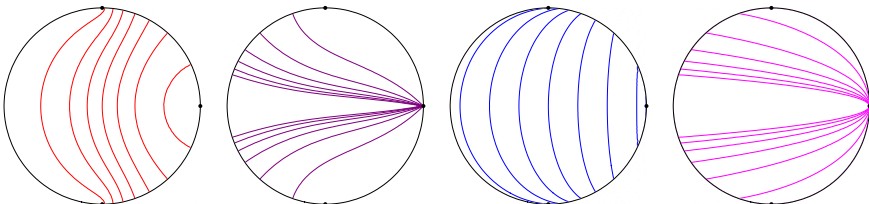

**Figure 12.** V. Wide and long concave hyperbolic parabolas. The left two parts show the horizontal (red) and vertical (magenta) trajectories in the Poincaré disc model; the right two parts show the horizontal (blue) and vertical (pink) trajectories in the Klein model.

(VI) Two double poles on the circle (Figure 13).
The quadratic differential is equivalent to

$$\frac{dz^2}{(z^2-1)^2}.$$

The trajectories consist of a hyperbolic straight line and the equidistant curves to the line (these being curves of constant curvature less than 1). The orthogonal trajectories are the geodesic field of lines orthogonal to the given line.

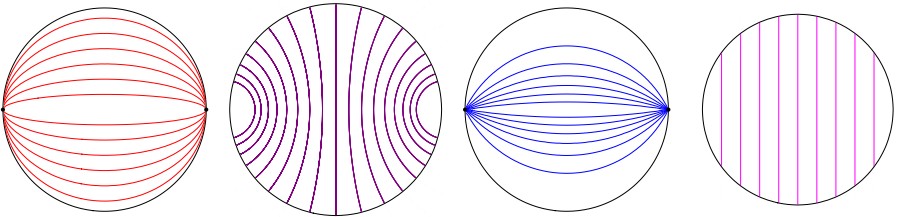

**Figure 13.** VI. Equidistant curves. The left two parts show the horizontal (red) and vertical (magenta) trajectories in the Poincaré disc model; the right two parts show the horizontal (blue) and vertical (pink) trajectories in the Klein model.

(VII) A double pole in the disc, and its reflected double pole (Figure 14).
The quadratic differential is equivalent to

$$\frac{dz^2}{z^2}.$$

The trajectories are concentric circles. If the center is chosen to be at the origin, the Euclidean, Poincaré, and Klein models are identical; otherwise, the curves in the Poincarè disc are circles, and the Klein circles are ellipses. The orthogonal trajectories are rays from the center.

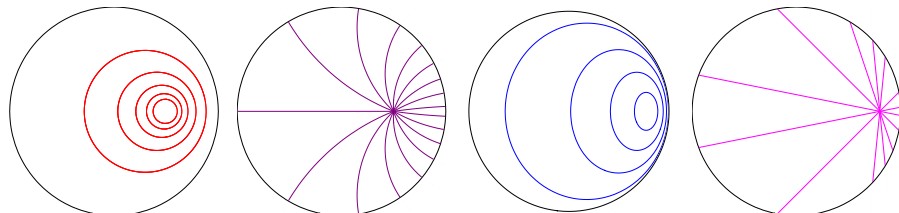

**Figure 14.** VII. Circles. The left two parts show the horizontal (red) and vertical (magenta) trajectories in the Poincaré disc model; the right two parts show the horizontal (blue) and vertical (pink) trajectories in the Klein model.

(VIII) A triple pole on the circle, and a single pole on the circle (Figure 15).
The quadratic differential is equivalent to

$$\frac{4dz^2}{(z+1)^3(z-1)}.$$

The trajectories are osculating parabolas, as are the orthogonal trajectories. The line $\ell$ joining the two foci reflects one of the two families to the other. Each parabola reflects rays orthogonal to $\ell$ to rays from the ideal point.

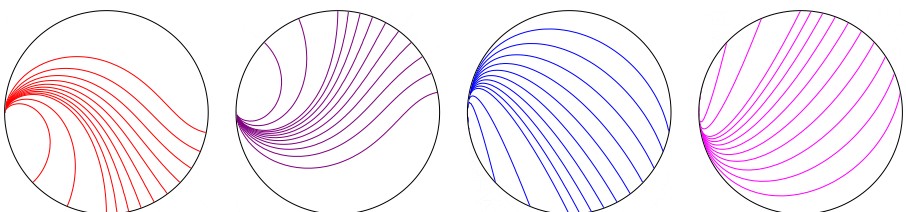

**Figure 15.** VIII. Osculating parabolas. The left two parts show the horizontal (red) and vertical (magenta) trajectories in the Poincaré disc model; the right two parts show the horizontal (blue) and vertical (pink) trajectories in the Klein model.

(IX) A quadruple pole on the circle (Figure 16). The quadratic differential is equivalent to

$$\frac{dz^2}{(z-1)^4}.$$

The trajectories are horocycles. The orthogonal trajectories are the lines meeting at the ideal point, i.e., the hyperbolic parallel lines. This is the limiting case of concentric circles, when the center becomes an ideal point.

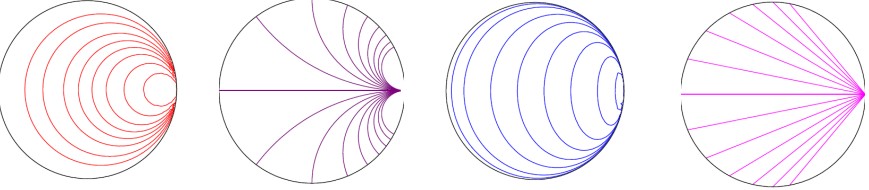

**Figure 16.** IX. Horocycles. The left two parts show the horizontal (red) and vertical (magenta) trajectories in the Poincaré disc model; the right two parts show the horizontal (blue) and vertical (pink) trajectories in the Klein model.

## 5. Conclusions

We have classified orthogonal pairs of meromorphic foliations of the Poincaré disc by hyperbolic conics. We showed that there are nine types of such foliations: three of them admit one parameter up to isometry of the hyperbolic plane; the remaining six types are unique up to isometry. Every hyperbolic conic is represented among the leaves of these foliations: this gives a complete classification of confocal families of hyperbolic conics, where the notion of confocal depends only on the geometry of the hyperbolic plane. There are other notions of foci that are model-dependent; while we use the Poincaré model to construct the foliations, the resulting families can be characterized in a model-independent way as mirrors reflecting one family of geodesic fields to another. Using the notion of the EPH field (elliptic–parabolic–hyperbolic), we are able to characterize confocal families, without dealing with the ambiguous notion of foci.

Figure 17 displays the nine families of horizontal and vertical trajectories in the Poincaré disc.

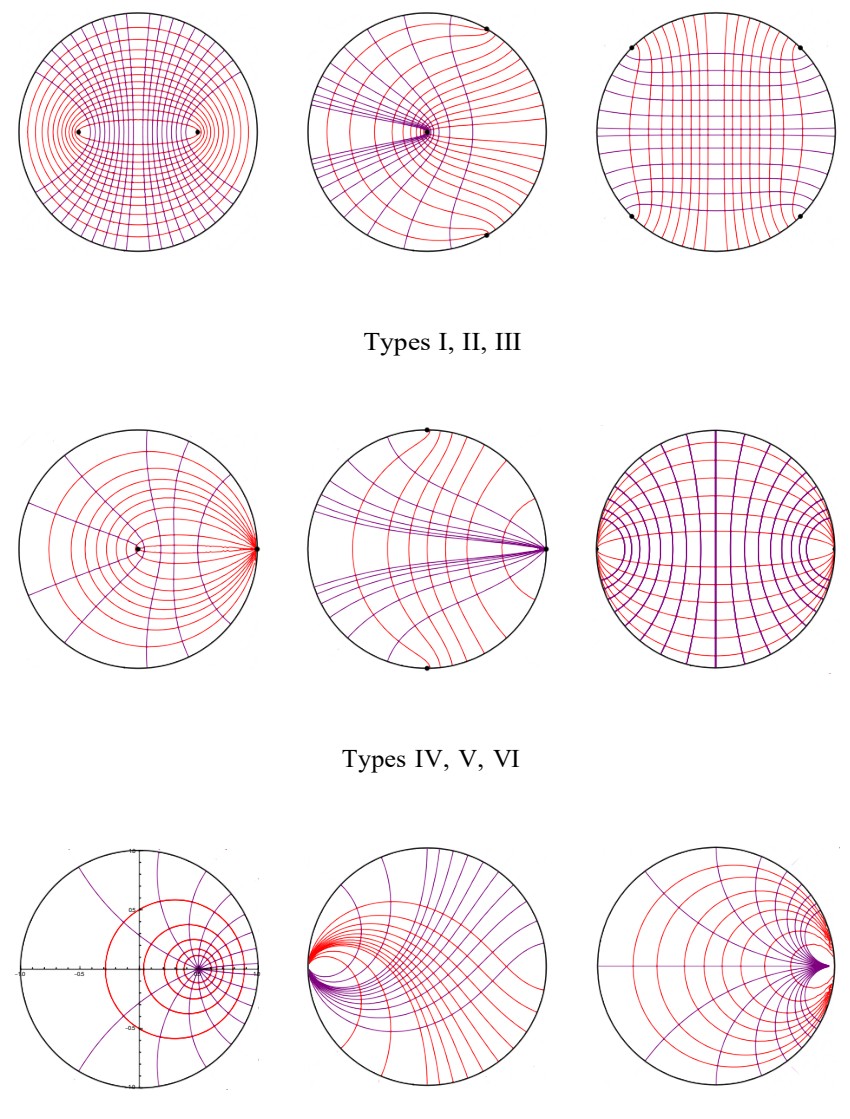

Types I, II, III

Types IV, V, VI

Types VII, VIII, IX

**Figure 17.** The nine orthogonal families of conics. The red curves are horizontal trajectories and the magenta curves are vertical trajectories.

**Author Contributions:** The two authors were equal partners in the conceptualization, formal analysis, visualization and writing of this project. J.L. took the lead in the use of *Mathematica* for producing graphics, while D.S. took the lead in the computational aspects. J.L. led in the writing, while D.S. led in the revision. All authors have read and agreed to the published version of the manuscript.

**Funding:** This research received no external funding.

**Data Availability Statement:** Not applicable.

**Conflicts of Interest:** The authors declare no conflict of interest.

## Appendix A. Conics and Their Names

Over the years, various authors have enumerated the types of hyperbolic conics. In an 1882 article, William Story listed 8 types, with some subtypes [6]. In Coolidge's 1909 textbook [2], he listed 11 types. Various modern authors have refined the list to 12 types; a particularly attractive treatment is a 2017 article by Ivan Izmestiev [4]. The various types

and their names are compiled in the following chart, with their correspondence to the confocal families:

**Table A1.** Comparative list of types of hyperbolic conics.

| Type | Story | Coolidge | Ismestiev |
|---|---|---|---|
| I | ellipse | ellipse | ellipse |
| I | hyperbola | convex hyperbola | convex hyperbola |
| II | semihyperbola | semihyperbola | semihyperbola |
| III | hyperbola | concave hyperbola | concave hyperbola |
| IV | elliptic parabola | elliptic parabola | elliptic parabola |
| IV | $\mathbb{H}$ parabola | convex $\mathbb{H}$ parabola | convex $\mathbb{H}$ parabola |
| V | $\mathbb{H}$ parabola | concave $\mathbb{H}$ parabola | wide/long parabola |
| VI | circle | equidistant curve | hypercycle |
| VII | circle | circle | circle |
| VIII | semicircular parabola | osculating parabola | osculating parabola |
| IX | circular parabola | horocycle | horocycle |

## Appendix B. Computation

The computations necessary to produce the curve families illustrated throughout this paper are accomplished exclusively using Mathematica. For example, the horizontal trajectories of the quadratic differential $\frac{dz^2}{\sqrt{(1-z^2)(1-k^2z^2)}}$ are given by the parametrized curves $f_c(t) = p\,\mathrm{sn}(t + ic, p^2)$, with $k = p^2$.

Using the formulas for addition of complex arguments of elliptic functions ([11], p. 24), we derive explicit coordinate functions $(x(t; p), y(t; p))$. Eliminating the parameter $t$, using the Mathematica command Eliminate, the curves can be expressed as the level sets of a family of quartic curves in the Cartesian coordinates $(x, y)$, depending on the parameter $p$.

Now, using a fractional linear transformation, we can map the poles to $(\pm a, \pm \frac{1}{a})$ to get the trajectories of Type I, or we can map the poles to four points on the unit circle, to get the trajectories of Type III. A similar technique, using elliptic cosine instead of elliptic sine, allows us to get the quartic equations corresponding to trajectories of Type II. The remaining six types do not involve elliptic integrals.

The corresponding figures in the Klein disc can be produced from the parameterized curves in the Poincaré disc, by composing with the Poincaré-to-Klein map, or by composing the Klein-to-Poincaré map, with the quartic equations for the curves in the Poincaré disc. The latter method reveals the Klein curves as Euclidean conics, which can then be extended beyond the disc.

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
