# Peer review of "Confocal Families of Hyperbolic Conics via Quadratic Differentials"

_axioms, doi:10.3390/axioms12060507_

Round 1
Reviewer 1 Report
Dear Editor
Please see the attached file

,
Reviewer 2 Report
See the attached PDF.

Author Response
I have made all of the nine corrections listed in the report. For item 8, I have deleted the reference.
Reviewer 3 Report
1. Brief summary of the manuscript.
In this article, the theory of quadratic differentials is applied to a classification of orthogonal pairs of foliations of the hyperbolic plane by means of hyperbolic conics. The article consists of four sections.
In Section 2 (Confocal conics in the hyperbolic plane), the general notion of confocal families of conics and of the bundles of light rays reflected by a confocal family is considered.
Section 3 (Curve families in the disc and meromorthic differentials) deals with the fields of geodesics and the corresponding differentials used in constructing the families.
In Section 4 (Classification of Confocal Families), the authors enumerate the quadratic differentials determined by the geodesic fields and nine types of confocal families of conics.
Appendix 1 gives a dictionary of the types of conics and Appendix 2 has some comments on computations.
2. The manuscript’s strengths.
The article is structured very well and represents profound and non-trivial results. The Introduction and References are complete and relevant.
The authors have presented a classification of orthogonal pairs of foliations of the hyperbolic plane by hyperbolic conics. In the article, the form of the quadratic differential whose horizontal and vertical trajectories give confocal families of conics and location of the poles are specified for each of the listed types.
All considered families are illustrated graphicaly in the Poincaré disc and in the Klein disc.
3. Provide a point-by-point list of your minor for the improvement of the manuscript.
There are no references to publication [3].
I think the manuscript can be accepted after minor revision. All my minor сorrections are in the attached file.

Author Response
I have made all the changes indicated in your report. I deleted what was reference 8. In the case of the reference numbered as 3, I had left off the citation to it. It is now referred to in the manuscript.
Round 2
Reviewer 1 Report
Dear Editor
The Paper is OK
.